

# Renyi entropy driven hierarchical graph clustering

Frédérique Oggier[1] and Anwitaman Datta[2]

[1] Division of Mathematical Sciences, Nanyang Technological University, Singapore, Singapore
[2] School of Computer Engineering, Nanyang Technological University, Singapore, Singapore

## ABSTRACT

This article explores a graph clustering method that is derived from an information theoretic method that clusters points in $\mathbb{R}^n$ relying on Renyi entropy, which involves computing the usual Euclidean distance between these points. Two view points are adopted: (1) the graph to be clustered is first embedded into $\mathbb{R}^d$ for some dimension $d$ so as to minimize the distortion of the embedding, then the resulting points are clustered, and (2) the graph is clustered directly, using as distance the shortest path distance for undirected graphs, and a variation of the Jaccard distance for directed graphs. In both cases, a hierarchical approach is adopted, where both the initial clustering and the agglomeration steps are computed using Renyi entropy derived evaluation functions. Numerical examples are provided to support the study, showing the consistency of both approaches (evaluated in terms of $F$-scores).

## INTRODUCTION

In this article we study the problem of graph clustering from an information theoretic perspective using within and between cluster Renyi entropy based estimators.

Clustering refers to an unsupervised learning task where, given a number of items, the goal is to group them in a "meaningful" way. It is unsupervised because one does not know a priori what makes items similar to each other, so as to justify that they belong to the same cluster. If we look at clustering from an information theoretic view point, items may be viewed as realizations of different random variables, and for two items to belong to the same cluster means that they are realizations instantiated from the same (or close enough) distributions (*Hartigan, 1975*), leading to consider distances among distributions, in particular the notion of mutual information. See for example *Sugiyama et al. (2014)*, *Faivishevsky & Goldberger (2010)*, *Steeg et al. (2014)*, *Wang & Sha (2011)* and *Müller, Nowozin & Lampert (2012)* for different ways to optimize the mutual information, yielding different forms of information theoretic clustering algorithms.

The idea of using Renyi entropy for clustering was originally proposed in the context of image processing (*Gokcay & Principe, 2002*; *Jenssen et al., 2003*). The underlying mechanism for the clustering uses two evaluation functions: one for within cluster evaluation and one for between cluster evaluation. The former minimizes the distance among probabilities in a given cluster, the latter maximizes the distance among clusters, both are motivated by a statistical distance which is derived from Renyi entropy. We provide the details for this in "Renyi Entropy Based Clustering Evaluation Functions".

Corresponding author
Anwitaman Datta,
anwitaman@ntu.edu.sg

The reason for choosing Renyi entropy vs another distance such as the mutual information is that to actually compute these information theoretic quantities, one would need to know the distribution of the data, which is unknown. This leads to the problem of finding a suitable estimator, a difficult problem of its own, while Renyi's entropy leads to an estimator which is linked to Gaussian kernels.

The motivation of this study is to explore whether and how that idea can be adapted in the context of graph clustering. This work is thus exploratory in nature, and focuses on identifying a new meaningful way to cluster graphs, rather than being driven by outperforming other graph clustering techniques (a task difficult to quantify in general, since there are often different valid ways to cluster a graph, giving different insights into its structure). In the case of image processing, the clustering task looks at grouping data points in a two-dimensional space, and the corresponding formulation of clustering using Renyi entropy captures the distance among groups of data points by considering them as instantiations of random variables, and thus infers differences in the probability distribution functions from which the points are drawn. Implicit in this model is the underlying distance metric of a coordinate space, which is used in the windowing process of the kernel density estimation (discussed in "Renyi Entropy Based Clustering Evaluation Functions"). Though the original works were intended primarily for images (thus, 2-dimensional spaces), the underlying distance metric has natural extensions to higher dimensional coordinate spaces, and thus adaptation of the clustering technique to data points in a higher dimension is also immediate.

In the context of graphs, clustering refers to grouping nodes according to the graph structure (typically in terms of edges or attributes, see e.g., *Schaeffer, 2007* for a survey). In fact, the family of graphs needs to be further discerned in the subcategories of directed and undirected graphs. When graphs are directed (meaning that edges have a direction), existing clustering techniques roughly fit two categories (*Malliaros & Vazirgiannis, 2013*): (i) techniques reducing the graph to an undirected graph, where the clustering is typically based on edge density, such as naive graph transformations (e.g., ignoring the directionality) or transformations maintaining directionality, and (ii) techniques designed specifically for directed graphs, typically based on patterns, such as citation-based or flow-based clusters.

As such, we identify two principal challenges, and addressing them comprise the contributions of this work, which we describe below.

(i) While undirected graphs have a notion of distance, namely the shortest path based distance, its role in the Renyi entropy based evaluation functions is not straightforward. We thus resort to the idea of embedding the $N$ graph vertices to $\mathbb{R}^N$. The term embedding here refers to a mapping $f$ that sends the $N$ graph vertices to $N$ points in $\mathbb{R}^N$, in such a way that the distances between the image by $f$ of the vertices are (ideally the same as the distances between the vertices themselves. This is typically not possible, in which case we look for an embedding that makes the distances between the image of the vertices as close as possible instead of being the same, introducing a distortion in the embedding. We embed graphs using a semidefinite programing approach that minimizes the distortion of the embedding, and then carry out clustering with respect to

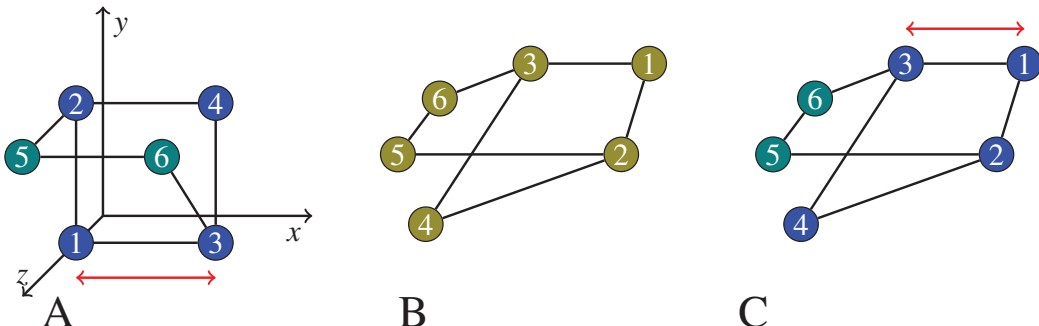

**Figure 1 In (B), we show a graph that we would like to cluster.** On (A), the graph vertices are first embedded into $\mathbb{R}^3$, trying to preserve their graph distances (for example, the square Euclidean distance between node 1 and node 3 is 1 on the left, which equals the shortest path distance between node 1 and 3 in the middle and on the right), but likely introducing certain distortions (the square Euclidean distance between node 4 and node 6 is 1 on the left, but there is no shortest path of length 1 in the middle and right graphs). Then the vertices now seen as points in $\mathbb{R}^3$ are clustered based on their 'new' distances in the coordinate space. On (C), the clustering is applied on the graph itself using the native graph distance measure (shortest path). Note that, the semidefinite programming approach we actually use would embed the 6 points in $\mathbb{R}^6$. But for the exposition of the principles of embedding and distortion, we show an embedding to $\mathbb{R}^3$.

the standard Euclidean norm to the resulting points in $\mathbb{R}^N$ using Renyi entropy based evaluation functions. The initial clusters are hierarchically agglomerated using the same evaluation functions.

In principle, the embedding is needed, since the kernel density estimation is based on a coordinate space assumption. We experimentally study the quality of embedded graph clusters, and observe that meaningful clusters (e.g., in terms of *F*-score, where groundtruth is known) are obtained despite the distortion introduced by the embedding process. However, beside inducing distortion, the embedding process is moreover computationally expensive. We thus then dispense the embedding, and investigate the clusters that are obtained when the graph's native (shortest path) distance metric is directly used which (does not induce any distortion but) is discordant with the coordinate space assumption of the kernel density estimator. Figure 1 depicts these two ideas. We compare experimentally the results against the clusters obtained with embedding and demonstrate that the quality of graph clustering is satisfactory for practical purposes even when we replace the Euclidean distance with a graph distance, and in the process avoid the computation intensive embedding process.

The practical algorithms for using Renyi entropy to create initial clusters and then carry out hierarchical agglomeration and accompanying experiment design and results to validate the efficacy of the two alternate approaches to reason about the use of the graph's native (shortest path) distance measure with the kernel density estimator are all reported in "Renyi Entropy Based Clustering of Undirected Graphs". We provide three types of experiments: (1) Comparison with ground truth in terms of *F*-score is provided for the dolphin network (*Lusseau et al., 2003*), (2) clustering of a subgraph of the Bitcoin network serves as an illustration of how the clustering performs on a larger network (it contains ~4,500 nodes), (3) clustering of a synthetically generated benchmark graph

with planted communities (*Lancichinetti, Fortunato & Radicchi, 2008*), where the communities are known by design.

(ii) The notion of distance in a directed graph is more subtle, and in particular, the standard concept of shortest path length between two nodes in an undirected graph does not satisfy the axioms of a metric. As such, our second challenge was to identify a proper metric, which is furthermore semantically meaningful from a graph clustering point of view. We consider being in/out-neighbors of the same nodes as a way to infer similarity among nodes (this aligns semantically with the existing accepted norm of (co-)citation based clustering of directed graphs), and demonstrate that a variation of the Jaccard similarity coefficient endows directed graphs with a distance metric. The previously designed clustering algorithms seamlessly work with this metric. We conduct experiments to evaluate the performance of clustering of directed graphs using it. We report these in "Renyi Entropy Based Directed Graph Clustering". Experiments are provided on synthetically created networks and the Montreal gang network (*Countinho, 2016*).

The above two contributions contrast with our preliminary work which first investigated the use of Renyi entropy for graph clustering (*Oggier, Phetsouvanh & Datta, 2018*), which was experiment centric, relied on simulated annealing (instead of hierarchical clustering, or the idea of embedding to establish the extent and effects of distortion in order to reason about the use of the graph's native distance metric in the kernel evaluation function), and did not consider directed graphs.

The article's organization is modular—the ideas and concepts are presented in individual self-contained sections, which also include the associated algorithmic details to implement the ideas and discussion of experiments and results validating those ideas. This narrative fits the exploratory nature of this work.

*Remark.* All the graph algorithms were implemented using Python and NetworkX (*Schult & Swart, 2008*). The semidefinite programs were solved using cvxopt (*Andersen, Dahl & Vandenberghe, 2013*) and cvxpy (*Diamond & Boyd, 2016*).

## RENYI ENTROPY BASED CLUSTERING EVALUATION FUNCTIONS

### Kernel density estimation

We start by recalling a well-known method to approximate a probability density function $p(\mathbf{x})$ given $N$ of its samples. Let $P = \int_{\mathscr{R}} p(\mathbf{x})d\mathbf{x}$ be the probability that a vector with pdf $p(\mathbf{x})$ falls in a region $\mathscr{R}$ of $\mathbb{R}^d$. If $\mathscr{R}$ is small enough that $p(\mathbf{x})$ varies little within it, $P$ can be approximated by

$$P \approx p(\mathbf{x}) \int_{\mathscr{R}} d\mathbf{x} \tag{1}$$

where $\int_{\mathscr{R}} dx$ is the volume of $\mathscr{R}$. Now given $N$ samples $\mathbf{x}_1,\ldots, \mathbf{x}_N$ which are independently drawn according to $p(\mathbf{x})$, say there are $k$ out of the $N$ samples falling within $\mathscr{R}$, then

$$P = \frac{k}{N} \tag{2}$$

Combining (1) and (2) gives $\hat{p}(\mathbf{x}) = \frac{k/N}{\int_{\mathscr{R}} d\mathbf{x}}$ as an immediate estimate for $p(\mathbf{x})$. If now $\mathscr{R}$ is a hypercube in $\mathbb{R}^d$ with edge length $h$ centered at $\mathbf{x}$, then $\int_{\mathscr{R}} d\mathbf{x} = h^d$, and the following window function $W$ indicates whether $\mathbf{x}_i$ is inside $\mathscr{R}$:

$$W_h(\mathbf{x}_i - \mathbf{x}) = \begin{cases} 1 & \text{if } \frac{|x_j - x_{ij}|}{h} \leq \frac{1}{2}, \ j = 1, \ldots, d \\ 0 & \text{else} \end{cases}$$

The total number $k$ of samples falling in $\mathscr{R}$ is thus given by $k = \sum_{i=1}^{N} W_h(\mathbf{x} - \mathbf{x}_i)$ and the Parzen window estimator (*Parzen, 1962*) is given by $\hat{p}(\mathbf{x}) = \frac{1}{N} \sum_{i=1}^{N} \frac{1}{h^d} W_h(\mathbf{x} - \mathbf{x}_i)$. More generally, the window function $W_h$ can be replaced by a symmetric multivariate probability density function, called kernel function, giving rise to a class of kernel density estimators. A typical choice for the window or kernel function $W_h$ is the Gaussian function given by $W_\sigma(\mathbf{x} - \mathbf{x}_i) = \frac{1}{\sqrt{2\pi^d}} \exp\left(-\frac{1}{2\sigma^2} ||\mathbf{x} - \mathbf{x}_i||^2\right)$, where $h = \sigma$ is a scale parameter. We thus get

$$\hat{p}(\mathbf{x}) = \frac{1}{N} \sum_{i=1}^{N} \frac{1}{\sigma^d} W_h(\mathbf{x} - \mathbf{x}_i) = \frac{1}{N} \sum_{i=1}^{N} \frac{1}{\sigma^d \sqrt{2\pi^d}} \exp\left(-\frac{1}{2\sigma^2} ||\mathbf{x} - \mathbf{x}_i||^2\right) \tag{3}$$

## A Renyi entropy sample-based estimator

The quadratic Renyi entropy of a vector $\mathbf{x} \in \mathbb{R}^d$ is defined by

$$H_2(p) = -\ln \int p^2(x) d\mathbf{x} \tag{4}$$

where $p(\mathbf{x})$ is the pdf of $\mathbf{x}$. When the pdf $p(\mathbf{x})$ is unknown, but samples are available, $p(\mathbf{x})$ can be replaced by a sample-based estimator in (4), to obtain an estimator of $H_2(p)$. Let us use the estimator (3):

$$\hat{H}_2(p) = -\ln \int \frac{1}{N^2} \sum_{i,j=1}^{N} \frac{1}{\sigma^d \sqrt{2\pi^d}} \exp\left(-\frac{||\mathbf{x} - \mathbf{x}_i||^2}{2\sigma^2}\right) \frac{1}{\sigma^d \sqrt{2\pi^d}} \exp\left(-\frac{||\mathbf{x} - \mathbf{x}_j||^2}{2\sigma^2}\right)$$

The product of two such Gaussian distributions is *Petersen & Pedersen (2012)*, 8.1.8

$$\frac{1}{\sigma^d \sqrt{2\pi^d}} \exp\left(-\frac{||\mathbf{x} - \mathbf{x}_i||^2}{2\sigma^2}\right) \frac{1}{\sigma^d \sqrt{2\pi^d}} \exp\left(-\frac{||\mathbf{x} - \mathbf{x}_j||^2}{2\sigma^2}\right)$$

$$= \frac{1}{\sqrt{(2\sigma^2)^d (2\pi)^d}} \exp\left(-\frac{||\mathbf{x}_i - \mathbf{x}_j||^2}{4\sigma^2}\right) \mathscr{N}_x\left(\frac{1}{2}(x_i + x_j), \frac{\sigma^2}{2}\mathbf{I}_d\right)$$

where $\mathcal{N}_x(\frac{1}{2}(\mathbf{x}_i + \mathbf{x}_j), \frac{\sigma^2}{2}\mathbf{I}_d)$ denotes a multivariate Gaussian distribution with mean $\frac{1}{2}(x_i + x_j)$ and covariance matrix $\frac{\sigma^2}{2}I_d$. Thus

$$\hat{H}_2(p) = \hat{H}_{2,\sigma^2}(p) = -\ln\frac{1}{N^2}\sum_{i,j=1}^{N}\frac{1}{(2\sigma)^d\sqrt{(2\pi)^d}}\exp\left(-\frac{||\mathbf{x}_i - \mathbf{x}_j||^2}{4\sigma^2}\right) \tag{5}$$

## Within-cluster and between-cluster evaluation functions

It is known (*Hartigan, 1975*) that one possible information theoretic formulation for the clustering problem is to suppose that each cluster corresponds to samples from a given probability distribution. Then separating the clusters becomes maximizing the distance[1] among distributions, while points within the same cluster should be close to each other, that is, the distance among points within the same cluster should be minimized.

The quantity $\hat{H}_2(p)$ in (5) was proposed in *Gokcay & Principe (2002)* and *Jenssen et al. (2003)* as a way to evaluate the distance within and between clusters: $\hat{H}_2(p)$ is interpreted as a *within-cluster* evaluation function (*Jenssen et al., 2003*), since if we consider a single cluster (with $N$ points), associated to a single pdf $p(\mathbf{x})$, $\hat{H}_2(p)$ computes an estimate of its entropy.

However, if we use (5) but by summing over two different clusters (there is a probability distribution $p_1(\mathbf{x})$, $p_2(\mathbf{x})$ associated to each cluster), then as proposed in *Gokcay & Principe (2002)*, *Jenssen et al. (2003)*, we get a *between-cluster* evaluation function

$$D_{\sigma^2}(\hat{p}_1, \hat{p}_2) = -\ln\frac{1}{N_1 N_2}\sum_{i=1}^{N_1}\sum_{j=1}^{N_2}\frac{1}{(2\sigma)^d\sqrt{(2\pi)^d}}\exp\left(-\frac{||\mathbf{x}_i - \mathbf{x}_j||^2}{4\sigma^2}\right)$$

as a cluster evaluation function that estimates the distance between two clusters, since the between-cluster entropy estimates the distance between sample-based estimation of their distributions. This also gives an estimator of the "cross Renyi entropy"—$\ln\int p_1(\mathbf{x})p_2(\mathbf{x})d\mathbf{x}$.

In the case of $C$ clusters with respective pdfs $p_1,\ldots,p_C$, we obtain the generalized *between-cluster* evaluation function

$$D_{\sigma^2}(\hat{p}_1,\ldots,\hat{p}_C) = -\ln\frac{1}{N^2}\sum_{i,j=1}^{N}\frac{\delta_{ij}}{(2\sigma)^d\sqrt{(2\pi)^d}}\exp\left(-\frac{||\mathbf{x}_i - \mathbf{x}_j||^2}{4\sigma^2}\right) \tag{6}$$

where $\delta_{ij} = 0$ if both $\mathbf{x}_i$ and $\mathbf{x}_j$ belong to the same cluster, and 1 otherwise. This function thus tries to globally separate the $C$ clusters from each other.

## RENYI ENTROPY BASED CLUSTERING OF UNDIRECTED GRAPHS

Consider now the problem of graph clustering. An undirected graph $G = (V, E)$ is defined by the set $V$ of its vertices, and the set $E$ of its edges. We assume that the graph is connected (otherwise, we consider the connected components of the graph separately). The graph

[1] The term distance is used loosely here, namely we do not necessarily mean a mathematical distance satisfying the 4 axioms of non-negativity, identity of indiscernibles, symmetry and triangle inequality.

$G = (V, E)$ forms a metric space together with the function $\rho : V \times V \rightarrow \mathbb{R}_{\geq 0}$ which associates to two vertices $u, v \in V$ the length of a shortest path between $u$ and $v$.

Indeed, $\rho$ satisfies the axioms of a distance: the length of a shortest path is always greater or equal to 0 (non-negativity), the shortest path of a point to itself has length 0 (identity of indiscernibles), the shortest path from $u$ to $v$ has the same length as that from $v$ to $u$ (symmetry), and the length of a shortest path satisfies the triangle inequality (going from $u$ to $v$ via $w$ is not shorter than going directly from $u$ to $v$).

In the "Within-Cluster and Between-Cluster Evaluation Functions", we saw how to define an evaluation function for clustering points in $\mathbb{R}^d$. The space $\mathbb{R}^d$ is equipped with zthe usual norm $l_2$, given by $||\mathbf{x}|| = \sqrt{\sum_{i=1}^{d} x_i^2}$, and the corresponding distance $||\mathbf{x} - \mathbf{y}||$ between $\mathbf{x}$ and $\mathbf{y}$.

Given these two metric spaces $(V, \rho)$ and $(\mathbb{R}^d, l_2)$, a function $f : V \rightarrow \mathbb{R}^d$ is called a $D$-embedding if for all $u, v \in V$,

$$r\rho(u, v) \leq ||f(u) - f(v)|| \leq Dr\rho(u, v)$$

where $r > 0$ is a scaling factor, $D$ indicates the distortion introduced by the embedding process. If $D = 1$, then we have an isometric embedding, and the problem of clustering $G$ with respect to $\rho$ is equivalent to clustering $f(V)$ in $\mathbb{R}^d$ with respect to the $l_2$ norm. However graphs which have a 1-embedding into $\mathbb{R}^d$ for some $d$ are rare. A universal result that gives a sense of the best that can be done for arbitrary graphs is found in the seminal work by *Bourgain (1985)* (the original work (*Bourgain, 1985*) used the $l_1$ norm), who proved that every $N$-point metric space $(X, \rho)$ (and thus in particular a graph with $N$ vertices) can be embedded into a Euclidean space with norm $l_2$ with a distortion of $O(\log N)$. This distortion is known to be tight, since some expander graphs are reaching it. There have been many works done later on around this topic, in particular to refine the embeddings (they are typically of probabilistic nature) and find the lowest dimension of the Euclidean space in which to embed $(X, \rho)$ (see e.g., the book by *Matousek (2002)*; Chapter 15) for a probabilistic embedding into a $c \log^2 N$-dimensional Euclidean space equipped with the $l_2$ norm). A computational embedding can be obtained using semidefinite programing (*Pataki, 2000*), for $d = N$, that is graph with $N$ vertices is embedded into $\mathbb{R}^N$.

We recall that a generic semidefinite program has the following form:

$$\min \quad tr(CX)$$

$$s.t. \operatorname{tr} \quad (A_i X) \leq b_i, i = 1, \ldots, m$$

$$X \succeq \mathbf{0}$$

where $X, C, A_i$, $i = 1, \ldots, m$ are all symmetric $N \times N$ matrices and $X \succeq \mathbf{0}$ means that $X$ is positive semidefinite (that is, satisfies that $\mathbf{a}^T X \mathbf{a} \geq 0$ for any column vector $\mathbf{a}$). We will show next how to choose $X, C, A_i$, $i = 1, \ldots, m$ so as to model a graph embedding problem. Consider the $N \times N$ matrix $F$ whose columns are the embeddings $f(v_1), \ldots, f(v_N)$ of the

graph vertices. Then the matrix $X = F^T F$ contains as coefficients $x_{ij} = \langle f(v_i), f(v_j) \rangle$ and is positive semidefinite. In order to have a $D$-embedding into $\mathbb{R}^N$, we need for all $v_i, v_j \in V$:

$$\rho(v_i, v_j)^2 \leq ||f(v_i) - f(v_j)||^2 \leq D^2 \rho(v_i, v_j)^2$$

$$\Leftrightarrow \rho(v_i, v_j)^2 \leq x_{ii} + x_{jj} - 2x_{ij} \leq D^2 \rho(v_i, v_j)^2$$

Therefore, for $\mathbf{d}$ a vector of norm $D$, that is such that $\langle \mathbf{d}, \mathbf{d} \rangle = D^2$, we can formulate the problem of finding a $D$-embedding into $\mathbb{R}^N$ which minimizes $D$ as the following semidefinite program:

min $\qquad\qquad\qquad\qquad \langle \mathbf{d}, \mathbf{d} \rangle$

s.t. $\qquad \rho(v_i, v_j)^2 \leq x_{ii} + x_{jj} - 2x_{ij} \leq D^2 \rho(v_i, v_j)^2, \ \forall i, j$

We note that this program optimizes $D$ given the dimension $N$ (therefore a higher dimensional embedding could give a smaller distortion).

Given an undirected graph $G = (V, E)$ to be clustered with respect to the shortest path length $\rho$, one could in principle compute the embedding that gives the smallest distortion $D$, and then compute the clustering of $f(V) \in \mathbb{R}^N$, which then becomes the problem of solving a noisy version of the original graph clustering problem, or vice-versa, the original graph clustering problem may be seen as a noisy version of the clustering in $\mathbb{R}^N$. If the noise is not too large, we may thus use the within and between-cluster evaluation functions with a distance $\rho$, the shortest distance between two nodes. If we wanted to embed an arbitrary graph with an arbitrarily large number of vertices, then the computational approach is not feasible, and we would have to rely on the theoretical bounds à la Bourgain to tell us what kind of distortion to expect. However our current approach is of hierarchical nature, and therefore we only need to invoke local graph embeddings to justify the initial clustering.

In the initial clustering of $G$ described by Algorithm 1, nodes that will serve as centers of their respective clusters are chosen uniformly at random, after which, neighbors are added around them, so as to keep the within-cluster evaluation function minimized. The search is performed locally, starting from the center and expanding in the neighborhood. We can thus take the center $u$ of the cluster, and consider the subgraph built by considering only nodes that are at given distance $\delta$ from it, that is, consider a ball $\mathscr{B}(u, \delta)$ around $u$ of radius $\delta$ with respect to $\rho$. For every cluster center $u_i$, we attach a ball $\mathscr{B}(u_i, \delta)$ such that the union $\cup_{i=1}^{C} \mathscr{B}(u_i, \delta)$ covers $V$, and in fact, the balls may have overlapping boundaries, so that the search algorithm can handle points at the boundaries. Each ball $\mathscr{B}$ of size $|\mathscr{B}|$ may then be embedded in $\mathbb{R}^{|\mathscr{B}|}$ which allows us to compute the distortion of the embedding. The actual clustering will not need the embedding computation anymore: considering local graph embeddings justify the use of the within-cluster evaluation function, by arguing that the distances involved are suitably approximated by distances in $\mathbb{R}^N$, with a demonstrably small distortion. We next

---

**Algorithm 1** Initial Clustering.

1: **procedure** INITIALCLUSTER $G = (V, E)$, $\sigma_1^2$, $C$ a number of initial clusters)

2:    Compute a shortest path between any $u, v \in V$.

3:    Choose $C$ nodes uniformly at random, assign one label to each.

        ▷ Each node is a starting point for a cluster.

4:    **while** some nodes are still unlabeled **do**

5:       **for** every cluster **do**

6:          Find neighbors of nodes in the given cluster.

7:          **if** there are unlabeled neighbors or labeled neighbors with a worse distance to another
cluster **then**

            ▷ Keep only the first condition for a fast run.

8:          Add the neighbor whose addition minimizes the within-cluster evaluation function $\hat{H}_{\sigma_1^2}$.

---

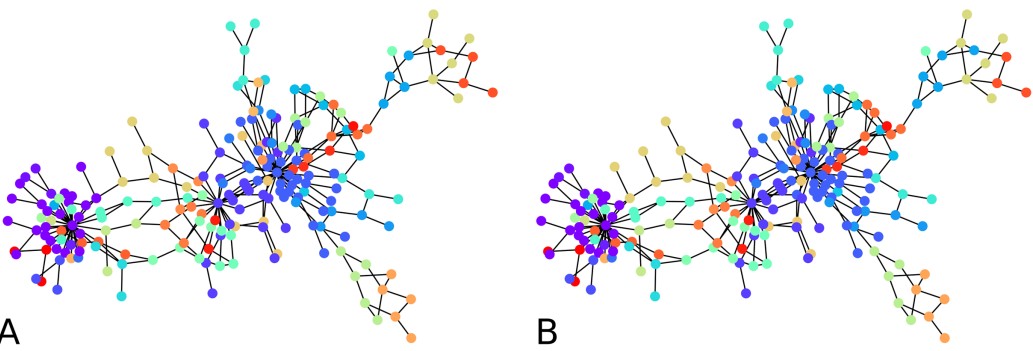

A                 B

**Figure 2 Output of the initial clustering of a Bitcoin subgraph with 209 nodes: (A) initial clustering of the embedded graph, (B) initial clustering of the graph as is.**

explore these ideas and the performance of the proposed graph clustering technique with experiments.

## Experiments

### *Exploration of embedding and distortion with a Bitcoin transaction subgraph*

Consider the graph shown on Fig. 2, which comprises 209 vertices. It is a subgraph representing connections induced by Bitcoin transactions among some wallet addresses. The graph data is found at *Oggier & Datta (2020)*. We repeatedly generated $C = 50$ random cluster centers $u_1, \ldots, u_C$, and computed $\mathscr{B}(u_i, 2)$ for each $i$. For each iteration of the process, we thus got graph embeddings of $\mathscr{B}(u_i, 2)$, 50 of them, whose statistics are summarized below:

| distortion $D$ | | size of local graphs | |
|---|---|---|---|
| max $D^2$ | 2.75603 | max $|\mathscr{B}|$ | 71 |
| average $D^2$ | 1.8 | average $|\mathscr{B}|$ | 23 |

The distortion remains low over all graph instances, the maximum distortion encountered was $D \approx 1.66$, and the average $D \approx 1.34$. The subgraph sizes varied up to 71 nodes, with an average size of 23 nodes. Note that to a point, the distortion depends more on the configuration of the vertices than on the size of the graph. Here in particular, the highest distortion happened for subgraphs of size 55, while the largest subgraphs had a distortion of $\approx \sqrt{2.65} \approx 1.62$. 1 means an embedding with no distortion.

We then ran our initial clustering algorithm twice, once on $V$ with distance $\rho$, and once on $\cup_{i=1}^{C} f(\mathscr{B}(u_i, 2))$ with distance $l_2$ computed over each embedding $f(\mathscr{B}(u_i, 2))$.

To evaluate the performance of a clustering which is not exact, we use the $F$-score (*Pfitzner, Leibbrandt & Powers, 2009*) of a clustering, which is defined as follows: a clustering is a partitioning of the nodes into subsets (the clusters), so let $C = \cup C_i$ be the correct partition of the nodes, and $C' = \cup C_j'$ be another clustering. For every $C_i \in C$, compute the set $P_2(C_i)$ of all subsets of two elements of $C_i$, and similarly the set $P_2(C_j')$ of all subsets of two elements of $C_j'$. Then for $P = \cup P_2(C_i)$ and $P' = \cup P_2(C_j')$, the $F$-score of $C'$ with respect to $C$ is defined by

$$\frac{2|P \cap P'|}{2|P \cap P'| + |P - P'| + |P' - P|} \tag{7}$$

An $F$-score of 1 means a perfect match. Thus, the closer to 1 the $F$-score, the more mutually similar the clusterings are.

We computed the $F$-score between the direct clustering of the graph, and the clustering of its embedding, and found an average $F$-score of $\approx 0.92$, which shows a high correlation between clusters obtained using the two approaches. Figure 2 visually compares the graph clustering with that of its embedding with an $F$-score of $\approx 0.87$. Even though it is hard to "see" a difference, across the 209 labels, 9 in fact disagree.

The initial clustering algorithm processes until all labels are assigned. It is usually run with the condition "if there are unlabeled neighbors or labeled neighbors with a worse distance to another cluster", but it could also be used with the simplified condition "there are unlabeled neighbors". The comparison shown in Fig. 2 between the clustering of the graph and that of its embedding in fact used the simplified condition. The latter makes the algorithm terminate fast. However, although more time-consuming, the condition that compares distances from one node to two clusters prevents one node to be assigned to a cluster without the possibility to be re-assigned to a better cluster. The algorithm terminates because at every iteration, either one non-assigned node gets a label, or an already assigned node changes label. Since the condition for changing label is determined based on a strict inequality, it is not possible for a node already assigned to infinitely move from one cluster to another.

Once an initial clustering is done, the next phase consists of an agglomerative clustering, as described in Algorithm 2. A cluster is drawn uniformly at random, the cluster closest to it (with respect to the in-between evaluation cluster $D_{\sigma_A^2}$) is identified and both of them are merged. Note that in our experiments we computed an averaging distance (one could

| **Algorithm 2** Agglomeration. |
| --- |
| 1: **procedure** AGGLOMERATION($G = (V, E)$, an initial clustering, $\sigma_A^2$) |
| 2:    Compute the between-cluster evaluation function between any two initial clusters. |
| 3:    **for** for every cluster from the initial clustering in a random order **do** |
| 4:        Find the closest initial cluster using the average between-cluster evaluation function $D_{\sigma_A^2}$. |
| 5:        Merge both clusters. |

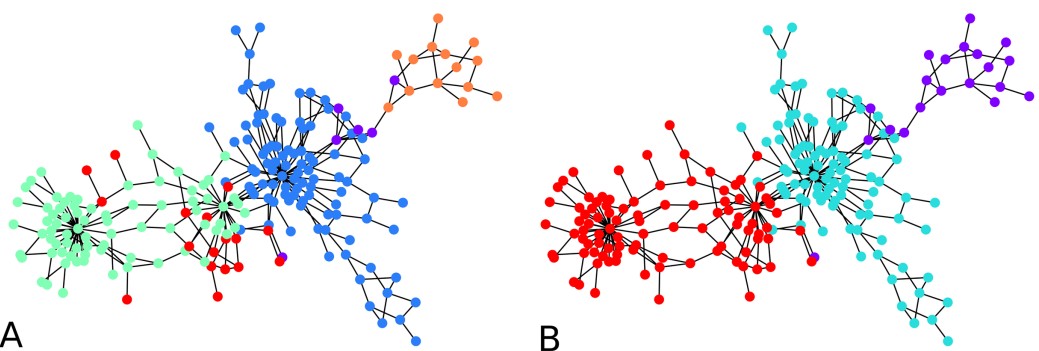

**Figure 3** An illustration of the hierarchical graph clustering algorithm on a 209 Bitcoin subgraph: (A) five clusters after the first round of agglomeration, (B) three clusters after the 2nd round of agglomeration.

choose a maximum or minimum instead, while retaining the rest of the framework). The process is iterated, yielding a bottom-up hierarchical clustering algorithm.

An illustration is provided in Fig. 3. The parameters chosen were $\sigma_I^2 = 0.01$ and $\sigma_A^2 = 1$ for every iteration of the agglomeration. There were 10 initial cluster centers, given by the nodes 38, 79, 75, 89, 149, 31, 185, 110, 190, 6. After the initial clustering, there were 10 clusters. After the first agglomeration, there were 5 clusters, which merged to form 3 clusters after second agglomeration. The clustering results for 5, respectively 3 clusters, are shown in Fig. 3. There is no ground truth attached to this graph and the result with 4 clusters shows three clusters around three high degree nodes, and a component on the right of the graph which has a minimum cut of 1 which forms a cluster of its own.

### The dolphin network

While the result from the experiments on the Bitcoin network seem reasonable visually, for further validation, we tried our clustering algorithm on the dolphin network (*Lusseau et al., 2003*) whose ground truth is known, that is, prior studies of this network have identified and accepted what are the communities. For clustering the dolphin network which comprises $N = 62$ nodes, we set $\sigma_I^2 = 0.01 \approx 1/N \approx 0.0161$, and $\sigma_A^2 = 1$ for every iteration of agglomeration. We were interested in an overall behavior of the algorithm, so we ran the algorithm with 10 nodes drawn uniformly at random as cluster centers. We look at the second round of agglomeration. This is because after the initial clustering, we have 10 clusters, and we expect roughly half of them, namely between 4 to

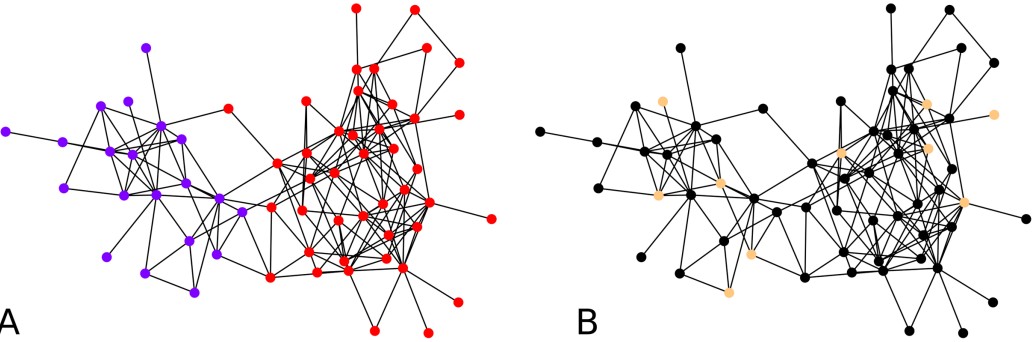

**Figure 4** Clustering of the dolphin network: (A) Groundtruth clustering of the dolphin network, (B) an example of initial choice of cluster centers (indicated in distinctive color) with which our algorithm results in a clustering identical to the groundtruth, that is, *F*-score = 1.

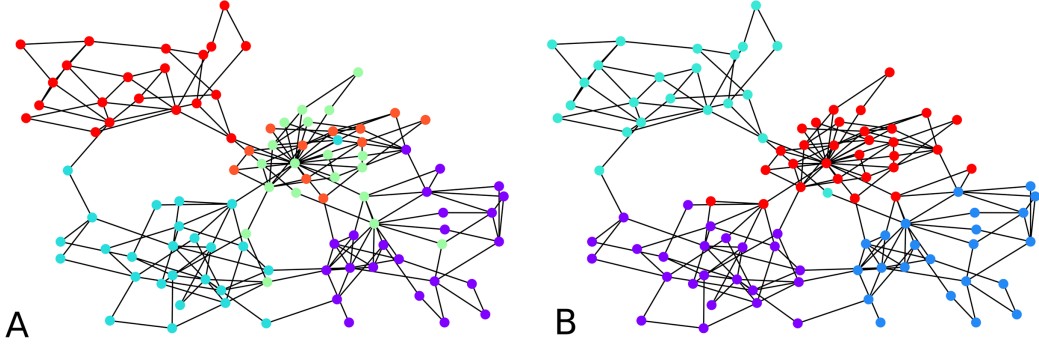

**Figure 5** Clustering of a synthetic (*Lancichinetti, Fortunato & Radicchi, 2008*) benchmark network: (A) *F*-score = 0.7693 (10 seed nodes, 1 round of agglomeration), (B) *F*-score = 0.8782 (20 seed nodes, two rounds of agglomeration).

6 clusters after the first round of agglomeration, and thus roughly 2 or 3 clusters after another round. Note that the *F*-score metric works even if the number of clusters in the two clusterings are different. Averaging over 1,000 iterations of the clustering algorithm results starting with random cluster centers as described above, each compared with the ground truth clusters, we obtained an average *F*-score of ≈ 0.7. If we look at the maximum *F*-score across these 1,000 experiments, we got an *F*-score of 1. An example of initial configuration that provides a perfect clustering is shown in Fig. 4B.

### Synthetic benchmark graph

In Fig. 5, clustering results are displayed for an instance of LFR graphs (*Lancichinetti, Fortunato & Radicchi, 2008*). This is a synthetic graph whose communities are planted. We thus know which and how many they are. In our example graph there are four communities. The left of the figure shows clusters obtained with one round of agglomeration when 10 initial points were chosen, and we used $\sigma_I^2 = 0.01$, giving an *F*-score of 0.7693 with respect to the ground truth. On the right, we show result from an experiment using 20 initial points and two rounds of agglomeration which resulted in an

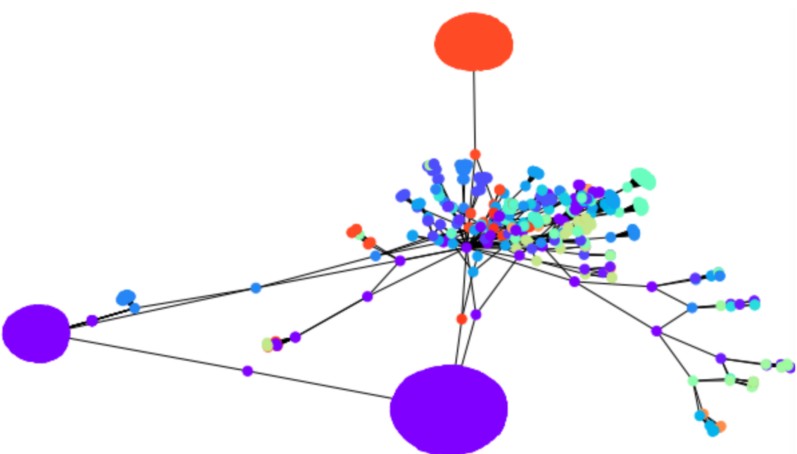

**Figure 6 Sixty-seven clusters of a Bitcoin subnetwork of 4,571 nodes.**

$F$-score of 0.8782. This clustering was consistent across different values of $\sigma_I^2$ (e.g., 0.02, 0.01 and 0.005).

**Larger scale experiment with a Bitcoin subgraph comprising 4,571 nodes**
To apply the algorithm on a relatively larger graph, we illustrate the initial clustering on a larger Bitcoin subgraph comprising 4,571 nodes. We drew 1,500 cluster centers uniformly at random, with $\sigma_I^2 = 0.01$, it resulted in 67 clusters, as shown in Fig. 6. There is no ground truth for this network, however, the clustering algorithm visually agrees with the drawing spring layout algorithm of NetworkX: nodes which are drawn together tend to be clustered together, with a number of clusters of 67 as a consequence of the original number of cluster centers (1,500 in this example). This is a meaningful observation because the mechanism behind the spring layout visualization of NetworkX is a force-directed algorithm (*Fruchterman & Reingold, 1991*), which itself can be used as a graph clustering algorithm (in the sense that the drawing algorithm is designed to preserve clusters, among other structural graph properties).

## RENYI ENTROPY BASED DIRECTED GRAPH CLUSTERING

We now consider the case of a directed graph $G = (V, E)$. A natural extension of the "Larger Scale Experiment with a Bitcoin Subgraph Comprising 4,571 nodes" would be to continue to use as a "distance" the length of a shortest path. However, in directed graphs, paths are directed, and there could be a path from $u$ to $v$, and yet no path from $v$ to $u$. Thus the length of the shortest path is not a distance metric anymore since it violates the axiom of symmetry. Instead, we are interested in a metric that will capture a (co-)citation like behavior: citation/co-citation consider two nodes close if they are often in/out-neighbors of the same nodes. In particular, we would like a metric that considers two nodes as close if both nodes are having many out-neighbors in common. For $u,v$ two nodes in $V$, denote by $\mathcal{N}_{\text{out}}(u)$ and $\mathcal{N}_{\text{out}}(v)$ the set of out-neighbors of $u$ and $v$ respectively. Then, assuming that $\mathcal{N}_{\text{out}}(u) \cup \mathcal{N}_{\text{out}}(v)$ is non-empty, the quantity

$$\frac{|\mathcal{N}_{\text{out}}(u) \cap \mathcal{N}_{\text{out}}(v)|}{|\mathcal{N}_{\text{out}}(u) \cup \mathcal{N}_{\text{out}}(v)|}$$

gives the fraction of common out-neighbors among all the out-neighbors. It varies from 0 to 1, and 1 happens exactly when all out-neighbors are common. This is known as the Jaccard similarity coefficient (attributed to *Jaccard (1912)*). The Jaccard distance

$$d_J(\mathcal{N}_{\text{out}}(u), \mathcal{N}_{\text{out}}(v)) = 1 - \frac{|\mathcal{N}_{\text{out}}(u) \cap \mathcal{N}_{\text{out}}(v)|}{|\mathcal{N}_{\text{out}}(u) \cup \mathcal{N}_{\text{out}}(v)|}$$

is a measure of dissimilarity, and it becomes 1 when no out-neighbors are common. We note that $d_J(u,v)$ does not exactly provide the measure we need, because two distinct nodes $u \neq v$ with the same set of neighbors would be at distance 0. Therefore we slightly modify the Jaccard distance for our purpose, and propose

$$\rho(u, v) = \frac{1}{2} (d_J(\mathcal{N}_{\text{out}}(u), \mathcal{N}_{\text{out}}(v)) + d_J(\mathcal{N}_{\text{out}}(u) \cup \{u\}, \mathcal{N}_{\text{out}}(v) \cup \{v\}))$$

**Lemma 1** The modified Jaccard distance $\rho$ satisfies the distance axioms.

*Proof.* **Identity of indiscernibles:** We have $\rho(u, v) \geq 0$ with equality if and only if $d_J(\mathcal{N}_{\text{out}}(u), \mathcal{N}_{\text{out}}(v)) = d_J(\mathcal{N}_{\text{out}}(u) \cup \{u\}, \mathcal{N}_{\text{out}}(v) \cup \{v\}) = 0$. It follows that $d_J(\mathcal{N}_{\text{out}}(u), \mathcal{N}_{\text{out}}(v)) = 0 \Leftrightarrow \mathcal{N}_{\text{out}}(u) = \mathcal{N}_{\text{out}}(v)$ and therefore $d_J(\mathcal{N}_{\text{out}}(u) \cup \{u\}, \mathcal{N}_{\text{out}}(v) \cup \{v\}) = 0 \Leftrightarrow u = v$.

**Symmetry** From the definition, we immediately have $\rho(u,v) = \rho(v,u)$.

**Triangle inequality**: We have

$$\rho(u, v) + \rho(v, w)$$
$$= \frac{1}{2} (d_J(\mathcal{N}_{\text{out}}(u), \mathcal{N}_{\text{out}}(v)) + d_J(\mathcal{N}_{\text{out}}(v), \mathcal{N}_{\text{out}}(w)) +$$
$$d_J(\mathcal{N}_{\text{out}}(u) \cup \{u\}, \mathcal{N}_{\text{out}}(v) \cup \{v\}) + d_J(\mathcal{N}_{\text{out}}(v) \cup \{v\}, \mathcal{N}_{\text{out}}(w) \cup \{w\}))$$
$$\geq \frac{1}{2} (d_J(\mathcal{N}_{\text{out}}(u), \mathcal{N}_{\text{out}}(w)) + d_J(\mathcal{N}_{\text{out}}(u) \cup \{u\}, black\mathcal{N}_{\text{out}}(w) \cup \{w\})) = \rho(u, w)$$

□

The normalization by 1/2 in the definition of $\rho$ just ensures that this quantity varies from 0 to 1. If now we have $u$ an out-leaf, that is a leaf such that $\mathcal{N}_{\text{out}}(u)$ is empty, and $v$ is another node with $\mathcal{N}_{\text{out}}(v)$ not empty, then there is no problem and $\rho(u, v) = 1$. However $\rho(u, u)$ cannot be computed and is set to 0. If both $u, v$ are out-leaves, we again cannot compute $\rho(u, v)$, and set the distance to be 1, the reason being that 1 is the largest distance, which is attributed to all pairs of nodes that do not have neighbors in common.

Equipped with this new distance metric in lieu of the shortest path length, we can apply the above algorithms for directed graphs. For Algorithm 1 the stop condition needs a minor adjustment for the condition "if some nodes are still unlabeled": indeed, it could now be possible to just have reached a stage where all clusters have no out-neighbor, in which case the algorithm should stop, and the algorithm is modified so that neighbors mean either out- or in-neighbors.

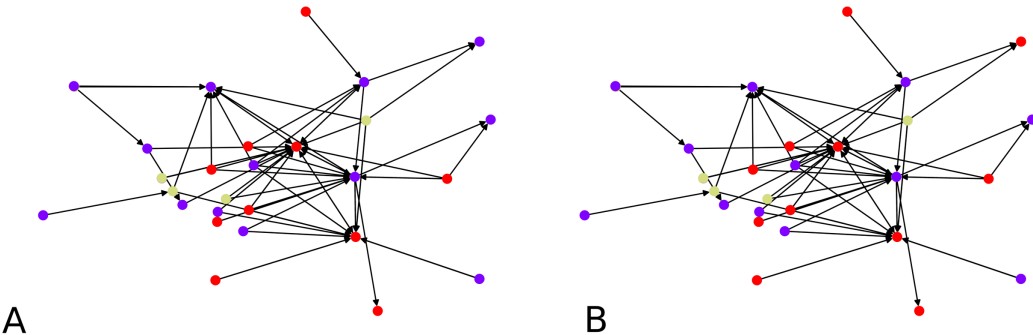

**Figure 7 Clustering of a 50 node directed scale-free graph and of its embedding: (A) Clustering of the embedded graph, (B) clustering of the graph directly.**

The discussion of the "Larger Scale Experiment with a Bitcoin Subgraph Comprising 4,571 Nodes" on graph embedding actually still holds here. The embedding of directed graphs is usually complicated, because directed graphs are asymmetric by nature. If one wanted to preserve the shortest path length say, it not being a distance (since the directed shortest path between $u$ and $v$ has no reason to be the same as that between $v$ and $u$), there would be no embedding with respect to the $l_2$ norm, though of course relaxations of distances could be considered (see e.g., *Linial, London & Rabinovich (1994)* for an isometric embedding using a "directed norm"). However, in our approach we actually consider a metric, namely $\rho$, and therefore an embedding with respect to $\rho$ is meaningful.

## Experiments
### Exploration with scale-free graphs
To validate our idea, we compute a local embedding using 4 balls of radius 0.7 according to $\rho$ on a random directed scale-free graph of 50 nodes. In Fig. 7 we show the two clustering instances (using the graph, and its embedding), which yield comparable clusters with $F$-score of $\approx 0.83$. The overall statistics for this graph instance, by choosing $C = 4$, are:

| distortion $D$ | | size of local graphs | |
|---|---|---|---|
| max $D^2$ | 1.14 | max $|\mathcal{B}|$ | 31 |
| average $D^2$ | 1.015 | average $|\mathcal{B}|$ | 17 |

We note that having an average subgraph size of 17 when sampling 4 subgraphs is telling that we should indeed be covering the graph by our local embeddings. The quality of the distortion indicates that our local embeddings are close to being (numerically) isometric though the computation of $F$-scores gives an average of $F$-score of $\approx 0.76$. The highest distortion is achieved by graphs of different sizes, such as 20, 26 or 31. It is possible to get a distortion of 0 when doing the local graph embedding if this graph comprises only one node, which obviously reduces the average.

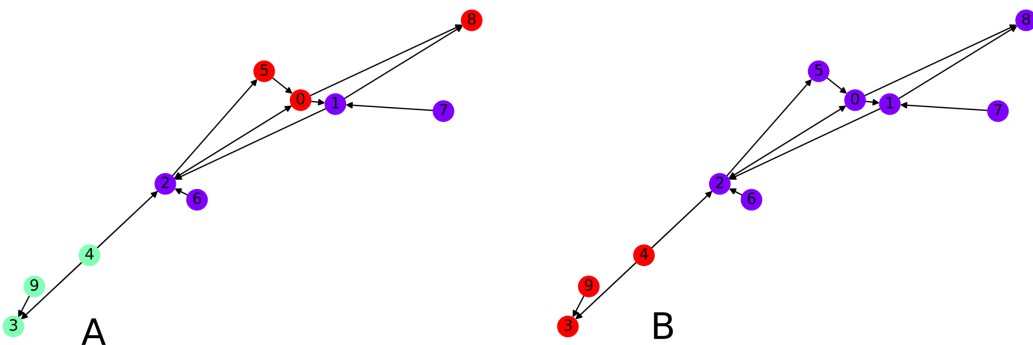

**Figure 8 Interpretation of the directed clustering: (A) A clustering of a random graph using 3 original cluster centers, 4, 6 and 8, as starting points, (B) a clustering of a random graph using 2 original cluster centers, 6 and 4, as starting points.**

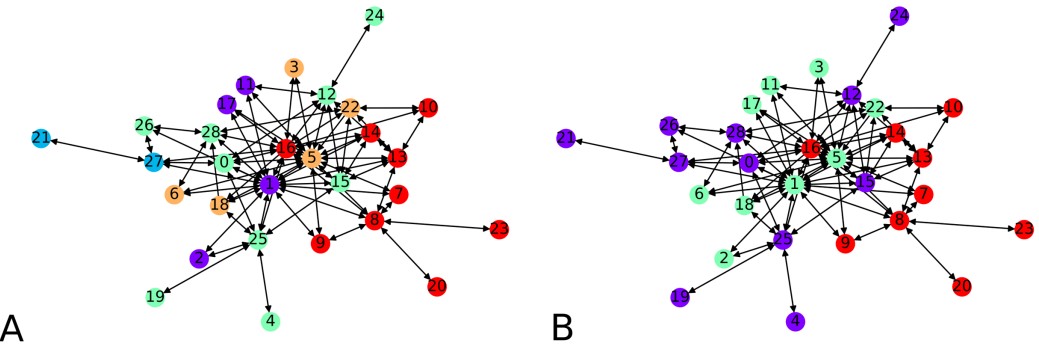

**Figure 9 Clustering of the Montreal Gang Network: (A) Five clusters after the initial clustering, cluster centers were 11, 21, 18, 7, 10, (B) three clusters after one round of agglomeration.**

Finally we focus on applying the clustering algorithm directly (without the intermediate of the graph embedding) to directed graphs. We apply our initial directed clustering algorithm on a toy example first, a random directed scale-free graph with 10 nodes. This allows us to explain the intuition behind the algorithm. On Fig. 8, nodes 9 and 4 are pointed at node 3, they form a cluster. Similarly nodes 6 and 1 are both pointed at node 2, so they form a cluster, together with node 2 which only points at node 1.

### Montreal gang network

We next apply our hierarchical clustering on the Montreal gang network (*Countinho, 2016*). The cluster centers are 5 points chosen uniformly at random. One instance outputs 5 clusters as shown in Fig. 9A. This is happening simply because the algorithm stopped after finding only clusters with no out-neighbor. Agglomerating these clusters gives the configuration showed on Fig. 9B. We notice three clusters, each comprising a few nodes which act as hubs, namely 16 and 8 in one group, 1, 5 and 22 in another, and 28 and 12 in the other one.

**Table 1 Summary of experiments and results.**

| anti-flashwhiteGraph | (un)directed | #nodes | embedded | direct | ground truth | *F*-score | Figure |
|---|---|---|---|---|---|---|---|
| Bitcoin | undirected | 209 | ✓ | × | no | ≈0.87 | 2 |
| | | | × | ✓ | no | | |
| Bitcoin | undirected | 209 | × | ✓ | no | × | 3 |
| Bitcoin | undirected | 209 | × | ✓ | no | × | 3 |
| Dolphin | undirected | 62 | × | × | yes | 1 | 4 |
| | | | × | ✓ | yes | | |
| LFR | undirected | 100 | × | ✓ | yes | ≈0.7693 | 5 |
| LFR | undirected | 100 | × | ✓ | yes | ≈0.8782 | 5 |
| Bitcoin | undirected | 4,571 | × | ✓ | no | × | 6 |
| Scale-free | directed | 50 | ✓ | × | no | ≈0.91 | 7 |
| | | | × | ✓ | no | | |
| Scale-free | directed | 10 | × | ✓ | no | × | 8 |
| Montreal | directed | 36 | × | ✓ | no | × | 9 |

## CONCLUSION

We proposed an exploratory study of the use of Renyi entropy based evaluation functions for the purpose of clustering both directed and undirected graphs, using two approaches: via graph embedding and via direct clustering on the graph data. The study serves as a proof of concept that with suitably defined distance metrics Renyi entropy based graph clustering is achievable for both directed and undirected graphs. Few graph clustering algorithms in the literature can deal with both undirected and directed graphs, while preserving the semantics of directionality for the latter. Our approach provides a unifying framework to do so, by creating an abstraction of node distance which is used in the kernel density estimator, while instantiating the actual distances using a suitable manner depending on the nature of the graph being directed or not. The abstraction of graph embedding and exploration of the impact of distortion from such embedding, and comparison of the resulting clusters with and without embedding helps demonstrate that even though the Renyi entropy based clustering approach we apply uses in principle a kernel density estimator which assumes an underlying coordinate space, replacing it with a graph based distance metric nevertheless yield meaningful (quantified using *F*-score measure) clusters in a computationally efficient manner since the computationally intensive semidefinite programing based embedding (studied to justify the approach) can be avoided in practice. To summarize, the exploration of these ideas help reason about and ground the practical graph clustering heuristics proposed in this paper within a mathematically robust conceptual framework. We validate these ideas with experiments using a wide range of real as well as synthetically generated graphs, with/out known groundtruth information about the community structures in these graphs. Table 1 provides a summary of our experiments and results. An extension of the proposed methods for the clustering of weighted graphs is an interesting future direction of research, which, if successful, would enhance the universality of the approach.

## ACKNOWLEDGEMENTS

The authors will like to thank Silivanxay Phetsouvanh for discussions on using Renyi entropy for graph clustering. The authors accordingly co-authored *Oggier, Phetsouvanh & Datta (2018)* with him, in which a very different approach than this paper, namely simulated annealing, is used.

### Funding

The authors received no funding for this work.

### Competing Interests

Anwitaman Datta is an Academic Editor for PeerJ.

### Author Contributions

- Frédérique Oggier conceived and designed the experiments, performed the experiments, analyzed the data, performed the computation work, prepared figures and/or tables, authored or reviewed drafts of the paper, and approved the final draft.
- Anwitaman Datta conceived and designed the experiments, analyzed the data, prepared figures and/or tables, authored or reviewed drafts of the paper, and approved the final draft.

### Data Availability

Data are available at Dataverse:

Oggier, Frederique Elise; Datta, Anwitaman, 2020, "A directed Bitcoin subgraph with 209 nodes", DOI 10.21979/N9/5CFO3I, DR-NTU (Data), V1.

Code is available in the Supplemental Files.

### Supplemental Information

Supplemental information for this article can be found online at http://dx.doi.org/10.7717/peerj-cs.366#supplemental-information.

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
