# Peer review of "Renyi entropy driven hierarchical graph clustering"

_PeerJ Computer Science, doi:10.7717/peerj-cs.366_

## Round 0.1 · original submission · Minor Revisions

Please provide point-to-point responses to the reviewers' comments. Thanks.

Reviewer 1 ·

Basic reporting

It is necessary to clearly state the purpose of the work. For example, the purpose of this paper is to apply the quadratic Renyi entropy to graph clustering. This type of entropy makes it possible to perform cluster analysis better in comparison with.... This hypothesis is proved by a series of computer experiments, which are described in the section.....

Experimental design

You need to add a separate section that describes all the experiments performed.

Validity of the findings

The conclusion does not fully reflect the results of the work.

Additional comments

In this paper, author consider the application of the quadratic Renyi entropy to estimate the intracluster and intercluster distance. The work is interesting and deserves publication. However, there are a number of unclear points in the work that need to be clarified.
1. the concept of ‘embeddings’ is actively used in the work (for example see row 54). But actually, embeddings’ usually associated with ‘word embeddings’ in machine learning. Author have to clarify that word or replace it by another word. The using of that word can leads to misunderstanding.
2. In General, it is not clear from the article what experiments were conducted. Therefore, you need to add a section that describes all the experiments. For example, in line 191-192, which describes a computational experiment on the dolphin network, the authors say that they used Sigma squared for 0.001 and 1. These were extreme values, or only two values were used? In other words, the description of all experiments must be clearly described, for example, to make a table of experiments.
3. in line 205, the authors say that the following ‘the clustering algorithm visually agrees: nodes which are drawn together tend to be clustered together..’. This is not clear. It is recommended to avoid such evaluation expressions, or explain them using the results of calculations.
4. in line 208, where there is a description of ' Jaccard distance’, you must provide a link, since this metric was not proposed by the authors of this work. However, in this paper it looks exactly like the metric proposed in this paper.
5. it is not clear from the paper whether the proposed approach was tested on datasets with a pre-known number of clusters. This needs to be clarified. If such experiments have not been performed, I recommend adding test results on datasets with a known number of clusters.
6. It is necessary to reformulate a conclusion. Finally, you need to add a summary of the results obtained for directed and undirected graphs.

Reviewer 2 ·

Basic reporting

Your introduction needs more detail. I suggest that you add a short literature review on the existing graph clustering techniques and their corresponding limitations. Thus, you could indicate the existing knowledge gap in the field. Moreover, I suggest that you add some comments on the explanation of your choice in favor of Renyi entropy among many types of entropy. Please add some comments on why you consider two approaches for clustering, namely, the reason to consider graph embeddings (some background knowledge and references are needed here).
Formal results include clear definitions of all terms and detailed proofs except lines 113-114, where the notations of X, C, A_i, b are not explained. Moreover, I suggest adding a reference to the result formulated in those lines (113-114).

Experimental design

I suggest that you formulate more precisely your research question, considering that you already explored the applicability of Renyi entropy for graph clustering in your previous work (Oggier, F., Phetsouvanh, S., and Datta, A. (2018)). The text from your Conclusion section could be used for the formulation of the research question.

Validity of the findings

no comment

Additional comments

The article is written in good, professional English and is well structured. I thank you for providing the source codes of the proposed clustering algorithms. Moreover, I commend the authors for a very detailed description of their theoretical methods and experiments. However, there are some typos and confusing phrases and notations in the work:

1) The phrase in lines 44-45 “we first embed the graph to be clustered into R^d using a semidefinite programming approach for a suitable dimension d that minimizes the distortion of the embedding” seems to be inconsistent with lines 115-116 where embeddings of graph vertices have dimension N. So, actually you do not vary the dimensionality d, you set it to be the number of vertices N. Also, the change of notations in line 117 concerning the dimensionality of embedding is confusing (everywhere above you write “D-embedding into R^d, but in line 117 you suddenly write D-embedding into R^N”). I suggest that you provide some comments or explanations about it.

2) I suggest implicitly write that D is the distortion, where notation D first appears (lines 101-102). In the same manner, I suggest to write what |B| means where it is introduced for the first time (namely, line 140).

3) Euclidean norm in line 100 is written incorrectly. Please, correct it.

4) Typo in the last line of the proof of triangle inequality (between lines 214 and 215): instead of N_{out}(v \union {w}) there must be N_{out}(w \union {w}). Please, correct it.

---

## Round 0.2 · accepted · Accept

The paper has been recommended to be acceptable. Please consider the reviewer's suggestions if appropriate when submit the final files.

Reviewer 1 ·

Basic reporting

The article is written clearly and does not require revision in this version. Literature review sufficient for the purposes of the article.

Experimental design

The article matches the scope of journal

Validity of the findings

The basic idea is that the density within a cluster is estimated using a Gaussian distribution. In general, it can be assumed that there may be datasets that do not have a Gaussian distribution. The authors showed dates with just such distributions. This is a limitation of both the theoretical and practical parts of this article.
However, exploring different types of densities is outside the scope of this article.

Additional comments

Now the articles become better. Therefore it can be published.
However, I would like to advise you to explore the possibility of constructing an estimate of the entropy not only using the Gaussian distribution but also other types of distributions.